

# Diagnostic value of serum procalcitonin, lactate, and high-sensitivity C-reactive protein for predicting bacteremia in adult patients in the emergency department

Chiung-Tsung Lin[1,2], Jang-Jih Lu[3,4], Yu-Ching Chen[1,5], Victor C. Kok[1,5,6] and Jorng-Tzong Horng[1,5,7]

[1] Department of Bioinformatics and Medical Engineering, Asia University Taiwan, Taichung, Taiwan
[2] Department of Laboratory Medicine, China Medical University Hospital, Taichung, Taiwan
[3] Department of Laboratory Medicine, Chang Gung Memorial Hospital at Linkou, Taoyuan, Taiwan
[4] Department of Medical Biotechnology and Laboratory Science, College of Medicine, Chang Gung University, Taoyuan, Taiwan
[5] Disease Informatics Research Group, Asia University Taiwan, Taichung, Taiwan
[6] Department of Internal Medicine, Kuang Tien General Hospital, Taichung, Taiwan
[7] Department of Computer Science and Information Engineering, National Central University, Taoyuan, Taiwan

Corresponding authors
Victor C. Kok, victorkok@asia.edu.tw
Jorng-Tzong Horng, horng@db.csie.ncu.edu.tw

## ABSTRACT

**Background.** Few studies compared the diagnostic value of procalcitonin with a combination of other tests including lactate and high-sensitivity C-reactive protein in the prediction of pathogenic bacteremia in emergency department adult patients.
**Methods.** We performed a retrospective study assessing the differences in performances of procalcitonin at a cutoff of 0.5 ng/mL, lactate at a cutoff of 19.8 mg/dL, high-sensitivity C-reactive protein at a cutoff of 0.8 mg/dL and their combinations for predicting bacteremia in emergency department adult patients. Sensitivity, specificity, overall accuracy, positive-test and negative-test likelihood, and diagnostic odds ratio with 95% confidence interval for each test combination were calculated for comparison. The receiver operating characteristic curve for every single test were compared using DeLong's method. We also performed a sensitivity analysis in two expanded patient cohorts to assess the discriminative ability of procalcitonin or test combination.
**Results.** A total of 886 patients formed the initial patient cohort. The area under the receiver operating characteristic curve for discriminating positive blood culture was: procalcitonin = 0.72 (95% CI [0.69–0.75]) with a derived optimal cutoff at 3.9 ng/mL; lactate 0.69 (0.66–0.72) with an optimal cutoff at 17.9 mg/dL; high-sensitivity C-reactive protein 0.56 (0.53–0.59) with an optimal cutoff of 13 mg/dL; with pairwise comparisons showing statistically significant better performance of either procalcitonin or lactate outperforming high-sensitivity C-reactive protein. To predict positive blood cultures, the diagnostic odds ratio for procalcitonin was 3.64 (95% CI [2.46–5.51]), lactate 2.93 (2.09–4.14), and high-sensitivity C-reactive protein 0.91 (0.55–1.55; $P = 0.79$). About combined tests, the diagnostic odds ratio for procalcitonin and lactate increases were 3.98 (95% CI [2.81–5.63]) for positive blood culture prediction. Elevated procalcitonin level rendered a six-fold increased risk of positive gram-negative bacteremia with a diagnostic odds ratio of 6.44 (95% CI [3.65–12.15]), which showed no further improvement in any test combinations. In the sensitivity analysis, as a single test

to predict unspecified, gram-negative and gram-positive bacteremia, procalcitonin performed even better in an expanded cohort of 2,234 adult patients in terms of the diagnostic odds ratio.

**Discussions**. For adult emergency patients, procalcitonin has an acceptable discriminative ability for bacterial blood culture and a better discriminative ability for gram-negative bacteremia when compared with lactate and high-sensitivity C-reactive protein. High-sensitivity C-reactive protein at a cutoff of 0.8 mg/dL performed poorly for the prediction of positive bacterial culture.

# INTRODUCTION

Bacterial bloodstream infection has been reported to have a rapid disease course, especially in patients admitted to the emergency department (ED) (*Lindvig et al., 2016*), and it is difficult to make an immediate and accurate diagnosis of bacteremia according to the clinical manifestations of patients. Sepsis is not easy to detect in the early phase, as the clinical manifestation could be latent or an exacerbation of a pre-existing condition, resulting in diagnostic difficulty (*Castelli et al., 2004*). Moreover, the mortality rate of blood-cultured patients in the medical ED has been reported to be high. In a cohort study, among patients who had blood cultures performed within 72 h of arrival to the medical ED, the overall 30-day mortality rate was 11% (*Lindvig et al., 2016*). The same study used a multivariate Cox model and demonstrated that bacteremia was one of the important prognostic factors of mortality among blood-cultured patients, with a hazard ratio of 1.4 (95% CI [1.1–1.8]). The mortality could be reduced by early detection and initiation of appropriate antibiotics. The ED of a hospital is the first point of entry for the majority of emergency patients. Distinguishing bacteremic sepsis from non-infectious systemic inflammation is very challenging, and blood culture is considered as the standard diagnostic approach (*Riedel, 2012*).

However, blood culture, including identification and drug sensitivity testing, requires at least 12–48 h; therefore, early-stage diagnosis is very important. In the past two decades, procalcitonin has been adopted for early-stage diagnosis. Procalcitonin is a precursor hormone of calcitonin. It was found that when the body is stimulated by an inflammatory response, especially bacterial infection, different cells in multiple organs secrete procalcitonin (*Linscheid et al., 2003*; *Nishikura, 1999*). Many studies have shown that procalcitonin has an excellent predictive ability for sepsis (*Arora et al., 2017*; *Kibe, Adams & Barlow, 2011*; *Nishikawa et al., 2017*). Additionally, another study showed that there is a risk of bacteremia in patients with acute fever when the procalcitonin level is greater than 0.5 ng/ml (*Kim et al., 2011*). However, few studies argued against the usefulness of procalcitonin for predicting bloodstream infection in certain clinical settings (*Aalto et al., 2004*; *Hoenigl et al., 2014*).

Clinically, the combination of the C-reactive protein level and white blood cell count is commonly used as a basis for determining infection (*Julian-Jimenez et al., 2015*; *Leli et al., 2014*; *Liu et al., 2017*; *Ljungstrom et al., 2017*), although there is a lack of solid evidence. C-reactive protein is a marker of acute inflammation and is associated with non-specific inflammatory responses by the human body to infection or trauma. Blood C-reactive protein levels rapidly rise during such events.

Several studies in the past have examined the role in the risk stratification or the discriminative ability of lactate in the management of the bacterial sepsis in emergency patients (*Freund et al., 2012*; *Ljungstrom et al., 2017*; *Shapiro et al., 2005*). However, few studies have focused on the discriminative power of elevated lactate levels in blood to predict bacteremia in the literature. The results of the tests for procalcitonin, C-reactive protein, and lactate can be obtained within 1 h, which may significantly shorten the time of decision-making for prescribing appropriate antibiotics if bacteremia is highly suggested by the results of the above tests, either singly or in combination.

This study focused on the differences in the performances of procalcitonin, lactate, high-sensitivity C-reactive protein (*Milone, Kamath & Israelite, 2014*; *Su et al., 2013*; *Windgassen et al., 2011*; *Yildiz et al., 2013*) and their combinations for predicting positive blood bacterial culture in adult patients in the ED and analyzed the discriminative ability of these tests to predict positive blood culture for any non-contaminant bacteria, gram-positive bacteria (GPB), and gram-negative bacteria (GNB) (*Nishikawa et al., 2016*; *Wang et al., 2016*). The test efficiency of single items was analyzed using the area under the receiver operating characteristic (ROC) curve (AUROC) (*Alemayehu & Zou, 2012*). The method with the greatest AUROC was used as a judgment standard for the relatively superior method among similar methods.

## MATERIALS & METHODS

### Collection of test data

This study used test data extracted from the dataset of a total of 41,358 blood bacterial culture bottle records (aerobic and anaerobic bottles were counted separately) at the ED of a medical center in central Taiwan between January 1, 2010 and December 31, 2010. In addition, the information system of the hospital was used to collect patient test reports, including 7,879 lactate test records, 50,287 high-sensitivity C-reactive protein test records, and 3,037 procalcitonin test records. In total, 886 records with all tests performed in 24 h were identified after examining the records.

### Research ethics and personal information protection

This study was approved by the Medical Research Department of China Medical University Hospital, Taichung, Taiwan (certificate number DMR-99-133). The study population included adult patients who were admitted to the Department of Emergency Medicine of China Medical University Hospital and underwent septic workup in 2010. After identifying the study candidates, data files were de-identified. The whole process of data mining and data manipulation strictly complied with the Taiwan Personal Information Protection Act. The study was classified as an exempt research study (number CMUH106-REC-003) by

the Research Ethics Committee of China Medical University Hospital, Taichung, Taiwan. The Ethics Committee also waived the need for written or verbal informed consent from the patients.

## Biological measurements

The medical center used UniCel DXC-800 (Beckman Coulter, Brea, CA, USA) to analyze high-sensitivity C-reactive protein, and a value of <0.8 mg/dl was used as the biological reference. Traditional C-reactive protein and high-sensitivity C-reactive protein involve the same protein in the plasma, and they have been shown to represent the same entity (*Windgassen et al., 2011*). Lactate measurements were also analyzed using DXC-800, with a biological reference range of 4.5–19.8 mg/dl. Procalcitonin testing was performed using a mini VIDAS (BioMérieux, Marcy-l'Étoile, France) B.R.A.H.M.S procalcitonin analysis (https://www.procalcitonin.com/pct-assays/pct-sensitive-kryptor.html; B.R.A.H.M.S., Hennigsdorf, Germany). The VIDAS B.R.A.H.M.S procalcitonin is an automated test for determining human procalcitonin in human serum or plasma (lithium heparin) using the enzyme-linked fluorescent assay technique. The biological reference is 0.5 ng/ml, and the functional sensitivity of the assay is 0.05 ng/ml. Laboratory standard operating procedures were used for plasma/serum analyses of the above three items. A total of two blood culture sets (each set consists of one aerobic bottle and one anaerobic bottle) for each patient was considered mandatory. The volume of blood drawn for each culture set was 20 ml. Bacterial blood culture and analysis were performed using BD BACTEC[TM] 9240 (Becton-Dickinson Diagnostic Instrument Systems, Baltimore, MD, USA). The BD BACTEC blood culture system is a fully automated microbiology growth and detection system designed to detect microbial growth in blood specimens.

Our research question was to examine the diagnostic value of elevated biomarkers using a cutoff at the upper limit of reference range (procalcitonin, 0.5 ng/mL; lactate, 19.8 mg/dL; and high-sensitivity C-reactive protein, 0.8 mg/dL) for predicting bacteremia, not to derive an optimal cutoff for each single test for the best discriminative power. However, the best cutoff would be presented where appropriate when a receiver operating characteristics curve analysis of a diagnostic biomarker was performed.

## Definition of true bacteremia

Blood culture is a standard method for determining bloodstream infections, and culture results can be as follows: (1) positive with a pathogen; (2) blood culture contamination, defined as either less than two blood cultures positive for coagulase-negative staphylococci or diphtheroids (catalase-positive gram-positive rods from multiple genera) (*Leal Jr, Jones & Gilligan, 2016*) or time to positivity for coagulase-negative staphylococci of more than 15 h (*Schuetz, Mueller & Trampuz, 2007*); and (3) negative without the culture of any bacteria from blood after more than 7 days.

True bacteremia was defined as growth of any significant, pathogenic bacterial species in one or more sets of blood cultures consisting of aerobic and anaerobic bottles. Common skin pathogens often considered as contaminants (namely, coagulase-negative Staphylococci, aerobic and anaerobic diphtheroids, Micrococcus species, Propionibacterium species,

**Table 1  Procalcitonin, high-sensitivity C-reactive protein (hs-CRP), and lactate levels versus blood culture results in 886 adult patients admitted to the emergency department, who underwent workup for presumed bacteremia or sepsis.**

| Test item | Blood culture results | | Total |
|---|---|---|---|
| | Positive with bacteria | Negative | |
| Procalcitonin level (ng/mL) | | | |
| ≥0.5 | 159 | 368 | 527 |
| <0.5 | 38 | 321 | 359 |
| Total | 197 | 689 | 886 |
| hs-CRP Level (mg/dL) | | | |
| ≥0.8 | 173 | 612 | 785 |
| <0.8 | 24 | 77 | 101 |
| Total | 197 | 689 | 886 |
| Lactate Level (mg/dL) | | | |
| ≥19.8 | 125 | 256 | 381 |
| <19.8 | 72 | 433 | 505 |
| Total | 197 | 689 | 886 |

**Notes.**
Cultures that regarded as a contaminant account for 3% (27/886) in this cohort of 886 patients. A statistically significant difference ($p < 0.0001$) was observed for procalcitonin levels according to patients' blood culture results (positive vs. negative). The blood culture result type (positive vs. negative) did not differ between patients with hs-CRP results ($p = 0.79$). A statistically significant difference ($p < 0.0001$) was observed for lactate results according to patients' blood culture results (positive vs. negative).

or Bacillus species) were excluded from this definition, except when the same species were isolated from at least two consecutive blood cultures (*Jaimes et al., 2004*; *Richter et al., 2002*). Mixed cultures were considered pathogenic when bacteria other than the contaminants were isolated.

## Statistical analysis

The study by *Riedel et al. (2011)* was used as a reference. Tables 1–4 show bacterial culture results and establishment of the testing methods. The pairwise chi-square test or Fisher's exact test was used for blood bacterial culture results (positive, contamination, or negative). The gold standard of sepsis diagnosis is a positive blood bacterial culture. ROC curves refer to the curves obtained from the results of various tests and bacterial blood cultures. In this study, ROC curves of GPC and GNB culture results and the three test items were obtained to investigate which single diagnostic method is the most superior method among all the diagnostic methods. For pairwise comparisons of two correlated ROC curves, we used DeLong's variance estimate to derive the standard error and identify the 95% CI (*DeLong, DeLong & Clarke-Pearson, 1988*).

In addition, we calculated the associations among all or some test values in single, paired, or all test items (procalcitonin, high-sensitivity C-reactive protein, and lactate), which were greater than the upper limit of the reference range, and blood bacterial culture results using traditional indicators, such as sensitivity, specificity, overall accuracy (equivalent to diagnostic effectiveness), positive-test likelihood, negative-test likelihood, preferred single indicator, and diagnostic odds ratio, to investigate the effectiveness of various test

**Table 2  Diagnostic variables associated with either single or combined tests for predicting positive blood culture.** Diagnostic variables associated with elevation of either procalcitonin (PCT ≥ 0.5 ng/ml), blood lactate (LAC ≥ 19.8 mg/dl), or high-sensitivity C-reactive protein (hs-CRP ≥ 0.8 mg/dl), two variables, and all variables for predicting positive blood culture.

| Pattern (n) | Sensitivity (95% confidence interval, CI) | Specificity (95% CI) | Overall accuracy (= Diagnostic effectiveness) (95% CI) | Positive-test likelihood (95% CI) | Negative-test likelihood (95% CI) | Odds ratio (95% CI) | P value |
|---|---|---|---|---|---|---|---|
| Elevated lactate (n = 381) | 0.63 (0.56–0.7) | 0.63 (0.59–0.66) | 0.63 (0.60–0.66) | 1.71 (1.47–1.96) | 0.58 (0.48–0.7) | 2.93 (2.09–4.14) | <0.0001 |
| Elevated PCT (n = 527) | 0.81 (0.74–0.86) | 0.47 (0.43–0.50) | 0.54 (0.51–0.57) | 1.51 (1.36–1.66) | 0.41 (0.31–0.55) | 3.64 (2.46–5.51) | <0.0001 |
| Elevated hs-CRP (n = 785) | 0.88 (0.82–0.92) | 0.11 (0.09–0.14) | 0.28 (0.25–0.31) | 0.99 (0.92–1.04) | 1.09 (0.71–1.66) | 0.91 (0.55–1.55) | 0.79 |
| Elevated lactate and PCT (n = 255) | 0.53 (0.46–0.6) | 0.78 (0.75–0.81) | 0.72 (0.69–0.75) | 2.41 (1.98–2.91) | 0.60 (0.51–0.70) | 3.98 (2.81–5.63) | <0.0001 |
| Elevated lactate and/or PCT (n = 653) | 0.91 (0.87–0.95) | 0.31 (0.28–0.35) | 0.45 (0.41–0.48) | 1.33 (1.24–1.42) | 0.28 (0.17–0.43) | 4.83 (2.84–8.69) | <0.0001 |
| Elevated hs-CRP and PCT (n = 497) | 0.76 (0.69–0.81) | 0.49 (0.46–0.53) | 0.55 (0.52–0.59) | 1.5 (1.34–1.66) | 0.49 (0.38–0.63) | 3.04 (2.1–4.44) | <0.0001 |
| Elevated hs-CRP and/or PCT (n = 815) | 0.93 (0.88–0.96) | 0.08 (0.06–0.11) | 0.27 (0.24–0.3) | 1.01 (0.96–1.05) | 0.86 (0.49–1.49) | 1.18 (0.63–2.34) | 0.70 |
| Elevated lactate and hs-CRP (n = 329) | 0.55 (0.48–0.62) | 0.68 (0.64–0.71) | 0.65 (0.62–0.68) | 1.71 (1.44–2.01) | 0.67 (0.56–0.78) | 2.57 (1.84–3.6) | <0.0001 |
| Elevated lactate and/or hs-CRP (n = 837) | 0.96 (0.93–0.99) | 0.06 (0.04–0.08) | 0.26 (0.23–0.29) | 1.03 (0.99–1.06) | 0.58 (0.27–1.24) | 1.76 (0.77–4.72) | 0.23 |
| Elevated lactate and PCT and hs-CRP (n = 234) | 0.48 (0.41–0.55) | 0.80 (0.77–0.83) | 0.73 (0.7–0.76) | 2.39 (1.94–2.93) | 0.65 (0.56–0.74) | 3.68 (2.59–5.22) | <0.0001 |

**Notes.**

$N$, 886; $n$, number of samples within the pattern.

Interpretation of results: negative-test likelihood <0.6 will indicate clinical usefulness.

Lin et al. (2017), *PeerJ*, DOI 10.7717/peerj.4094

**Table 3 Diagnostic variables associated with elevation of either PCT, LAC, or hs-CRP, two variables, and all variables for predicting blood culture positive for GNB.** Diagnostic variables associated with elevation of either procalcitonin (PCT $\geq$ 0.5 ng/ml), blood lactate (LAC $\geq$ 19.8 mg/dl), or high-sensitivity C-reactive protein (hs-CRP $\geq$ 0.8 mg/dl), two variables, and all variables for predicting blood culture positive for gram-negative bacteria.

| Pattern (n) | Sensitivity (95% Confidence Interval, CI) | Specificity (95% CI) | Overall accuracy (= Diagnostic effectiveness) (95% CI) | Positive-test likelihood (95% CI) | Negative-test likelihood (95% CI) | Odds ratio (95% CI) | P value |
|---|---|---|---|---|---|---|---|
| Elevated lactate (n = 337) | 0.64 (0.55–0.73) | 0.63 (0.59–0.66) | 0.63 (0.6–0.66) | 1.73 (1.46–2.02) | 0.57 (0.44–0.71) | 3.04 (2.02–4.63) | <0.0001 |
| Elevated PCT (n = 479) | 0.88 (0.81–0.93) | 0.47 (0.43 –0.50) | 0.53 (0.5–0.56) | 1.65 (1.49–1.81) | 0.26 (0.16–0.40) | 6.44 (3.65–12.15) | <0.0001 |
| Elevated hs-CRP (n = 726) | 0.90 (0.84–0.95) | 0.11 (0.09–0.14) | 0.23 (0.21–0.26) | 1.02 (0.94–1.07) | 0.85 (0.48–1.49) | 1.20 (0.62–2.49) | 0.70 |
| Elevated lactate and PCT (n = 224) | 0.58 (0.49–0.67) | 0.78 (0.75–0.81) | 0.75 (0.72–0.78) | 2.64 (2.14–3.22) | 0.54 (0.43–0.65) | 4.9 (3.24–7.45) | <0.0001 |
| Elevated lactate and/or PCT (n = 592) | 0.94 (0.89–0.98) | 0.31 (0.28–0.35) | 0.41 (0.38–0.44) | 1.38 (1.28–1.47) | 0.18 (0.09–0.35) | 7.75 (3.56–20.03) | <0.0001 |
| Elevated hs-CRP and PCT (n = 453) | 0.83 (0.76–0.89) | 0.49 (0.46–0.53) | 0.55 (0.51–0.58) | 1.65 (1.47–1.83) | 0.34 (0.22–0.49) | 4.89 (2.96–8.43) | <0.0001 |
| Elevated hs-CRP and/or PCT (n = 752) | 0.95 (0.90–0.98) | 0.08 (0.06–0.11) | 0.22 (0.19–0.25) | 1.04 (0.98–1.08) | 0.58 (0.26–1.26) | 1.8 (0.76–5.23) | 0.24 |
| Elevated lactate and hs-CRP (n = 294) | 0.58 (0.49–0.67) | 0.68 (0.64–0.71) | 0.66 (0.63–0.7) | 1.81 (1.49–2.15) | 0.62 (0.49–0.75) | 2.91 (1.94–4.39) | <0.0001 |
| Elevated lactate and/or hs-CRP (n = 769) | 0.97 (0.92–0.99) | 0.06 (0.04–0.08) | 0.2 (0.17–0.23) | 1.03 (0.98–1.06) | 0.52 (0.2–1.35) | 1.99 (0.70–7.74) | 0.27 |
| Elevated lactate and PCT and hs-CRP (n = 206) | 0.53 (0.44–0.62) | 0.80 (0.77–0.83) | 0.76 (0.73–0.79) | 2.64 (2.1–3.27) | 0.59 (0.48–0.7) | 4.48 (2.96–6.81) | <0.0001 |

**Notes.**

*N*, 886; *n*, number of samples within the pattern.

Lin et al. (2017), *PeerJ*, DOI 10.7717/peerj.4094

**Table 4  Diagnostic variables associated with elevation of either PCT, LAC, or hs-CRP, two variables, and all variables for predicting blood culture positive for GPB.**
Diagnostic variables associated with elevation of either procalcitonin (PCT ≥ 0.5 ng/ml), blood lactate (LAC ≥ 19.8 mg/dl), or high-sensitivity C-reactive protein (hs-CRP ≥ 0.8 mg/dl), two variables, and all variables for predicting blood culture positive for gram-positive bacteria.

| Pattern (n) | Sensitivity (95% Confidence Interval, CI) | Specificity (95% CI) | Overall Accuracy (= Diagnostic effectiveness) (95% CI) | Positive-test likelihood (95% CI) | Negative-test likelihood (95% CI) | Odds ratio (95% CI) | P value |
|---|---|---|---|---|---|---|---|
| Elevated lactate (n = 302) | 0.63 (0.51–0.74) | 0.63 (0.6–0.66) | 0.63 (0.6–0.66) | 1.7 (1.36–2.04) | 0.59 (0.42–0.78) | 2.88 (1.7–4.94) | <0.0001 |
| Elevated PCT (n = 418) | 0.68 (0.57–0.79) | 0.47 (0.43–0.50) | 0.49 (0.45–0.52) | 1.28 (1.06–1.49) | 0.68 (0.47–0.93) | 1.89 (1.11–3.33) | 0.02 |
| Elevated hs-CRP (n = 673) | 0.84 (0.73–0.91) | 0.11 (0.09–0.14) | 0.18 (0.15–0.21) | 0.94 (0.82–1.02) | 1.47 (0.83–2.49) | 0.64 (0.32–1.37) | 0.25 |
| Elevated lactate and PCT (n = 184) | 0.45 (0.34–0.57) | 0.78 (0.75–0.81) | 0.75 (0.72–0.78) | 2.06 (1.52–2.7) | 0.70 (0.55–0.85) | 2.93 (1.73–4.96) | <0.0001 |
| Elevated lactate and/or PCT (n = 536) | 0.86 (0.76–0.93) | 0.31 (0.28–0.35) | 0.37 (0.33–0.4) | 1.26 (1.11–1.37) | 0.44 (0.24–0.76) | 2.87 (1.43–6.41) | 0.003 |
| Elevated hs-CRP and PCT (n = 394) | 0.63 (0.51–0.74) | 0.49 (0.46–0.53) | 0.51 (0.47–0.54) | 1.25 (1.01–1.48) | 0.75 (0.54–0.99) | 1.67 (0.99–2.86) | 0.06 |
| Elevated hs–CRP and/or PCT (n = 697) | 0.89 (0.80–0.95) | 0.08 (0.06–0.11) | 0.16 (0.13–0.19) | 0.97 (0.87–1.03) | 1.32 (0.66–2.56) | 0.73 (0.33–1.86) | 0.57 |
| Elevated lactate and hs-CRP (n = 258) | 0.51 (0.39–0.63) | 0.68 (0.64–0.71) | 0.66 (0.63–0.7) | 1.58 (1.21–1.99) | 0.73 (0.56–0.9) | 2.17 (1.3–3.65) | 0.002 |
| Elevated lactate and/or hs–CRP (n = 717) | 0.96 (0.88–0.99) | 0.06 (0.04–0.08) | 0.15 (0.12–0.17) | 1.02 (0.94–1.06) | 0.67 (0.22–1.95) | 1.51 (0.46–7.83) | 0.67 |
| Elevated lactate and PCT and hs–CRP (n = 169) | 0.41 (0.3–0.53) | 0.80 (0.77–0.83) | 0.76 (0.73–0.79) | 2.04 (1.46–2.72) | 0.74 (0.59–0.87) | 2.76 (1.61–4.68) | <0.0001 |

**Notes.**

N, 886; n, number of samples within the pattern.

combinations. A two-sided *P*-value <0.05 was considered statistically significant. Statistical analyses were performed using the StatsDirect statistical software.

We also performed a sensitivity analysis that assessed the discriminative ability of a test or test combination in an expanded cohort of patients from the original dataset, in whom only the procalcitonin test or two tests (procalcitonin + lactate) had been performed in addition to standard blood culture. The diagnostic performance results from these two expanded cohorts of patients are displayed along with the initial research cohort of 886 patients for comparison purposes. Since the group sizes are important for the test performance, the prevalence of positive blood culture in the two expanded cohorts are as follows: the expanded cohort that consisted of 923 patients who had procalcitonin, lactate and blood cultures taken on the same day had a prevalence of true bacteremia of 21.9% (202+/923). Another expanded cohort of 2,234 patients who had procalcitonin test result and blood culture results had a prevalence of true bacteremia at 15.5% (347+/2,234).

## RESULTS

Between January 2010 and December 2010, a total of 886 adult patients admitted to the ED underwent all three tests (procalcitonin, lactate, and high-sensitivity C-reactive protein) and at least two sets of blood culture within 24 h of the workup window. In this ED cohort, 22.2% (197/886) of the patients showed positive results on blood culture. Blood cultures that had contaminants were noted in 3.05% (27/886) of the patients (Table 1). The cut-off values selected for the three tests were above the upper limit of the normal range ($\geq$0.5 ng/ml for procalcitonin, $\geq$19.8 mg/dl for lactate, and $\geq$0.8 mg/dl for high-sensitivity C-reactive protein).

We performed ROC curve analyses for every single test, procalcitonin, high-sensitivity C-reactive protein, and lactate in all 886 adult patients to compare their discriminative power in the prediction of the blood culture positive for bacteria (Fig. 1), GNB (Fig. 2), and GPB (Fig. 3). A Wilcoxon estimate of the AUROC revealed the following results for the discriminative ability in positive blood culture prediction: procalcitonin = 0.72 (95% CI [0.69–0.75]) with the best cutoff at 3.9 ng/mL; lactate = 0.69 (95% CI [0.66–0.72]) with the best cutoff at 17.9 ng/dL; high-sensitivity C-reactive protein = 0.56 (95% CI [0.53–0.59]) with the best cutoff at 13 mg/dL. Pairwise comparisons of ROC curves using DeLong's methodology (*DeLong, DeLong & Clarke-Pearson, 1988*) showed that the difference in the discriminative ability between either procalcitonin ($P < 0.001$) or lactate ($P < 0.001$) and high-sensitivity C-reactive protein was statistically significant. There was no difference between procalcitonin and lactate ($P = 0.30$) (Fig. 1).

For GNB prediction, the Wilcoxon estimate of the AUROC revealed the following results: procalcitonin = 0.79 (95% CI [0.76–0.81]) with the best cutoff at 3.9 ng/mL; lactate = 0.71 (95% CI [0.68–0.74]) with the best cutoff at 25.1 mg/dL; high-sensitivity C-reactive protein = 0.60 (95% CI [0.56–0.63]) with the best cutoff at 4.42 mg/dL. Pairwise comparisons of ROC curves showed that procalcitonin was statistically better than lactate ($P = 0.01$) and high-sensitivity C-reactive protein ($P < 0.001$), while lactate was statistically better than high-sensitivity C-reactive protein ($P = 0.003$) (Fig. 2).

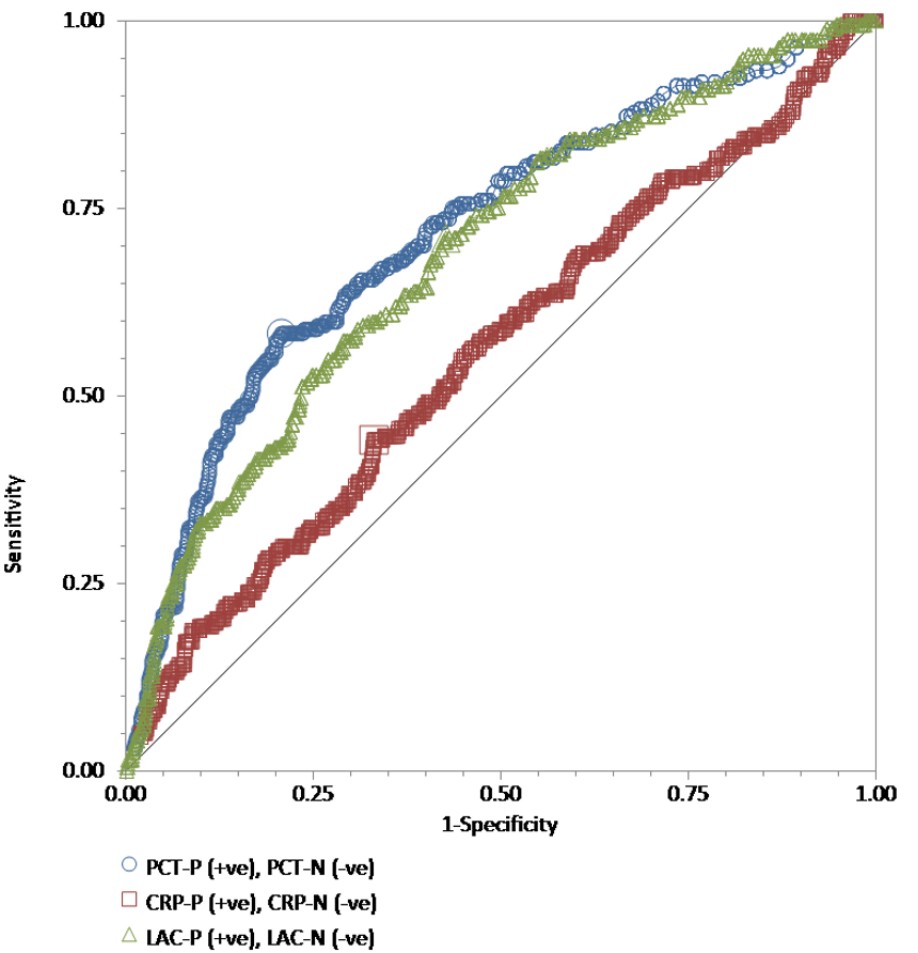

**Figure 1  The ROC curves for procalcitonin, high-sensitivity C-reactive protein, and lactate results relative to the "gold standard" of blood culture positive for bacteria.** The receiver operating characteristic (ROC) curves for procalcitonin (PCT), high-sensitivity C-reactive protein (hs-CRP), and lactate (LAC) results relative to the "gold standard" of blood culture positive for bacteria in a total of 886 adult patients admitted to the emergency department. Wilcoxon estimates of the area under the ROC curve are as follows: PCT = 0.72 (95% CI [0.69–0.75]); LAC = 0.69 (95% CI [0.66–0.72]); CRP = 0.56 (95% CI [0.53–0.59]). Pairwise comparisons of ROC curves show that the difference between either PCT ($P < 0.001$) or LAC ($P = 0.001$) and hs-CRP is statistically significant. There is no difference between PCT and LAC ($P = 0.30$).

For GPB prediction, the Wilcoxon estimate of the AUROC revealed the following results: procalcitonin = 0.61 (95% CI [0.57–0.64]) with the best cutoff at 4.13 ng/mL; lactate = 0.66 (95% CI [0.63–0.70]) with the best cutoff at 18.7 mg/dL; high-sensitivity C-reactive protein = 0.50 (95% CI [0.46–0.54]) with the best cutoff at 13.18 mg/dL. Pairwise comparisons of ROC curves showed that the difference between either procalcitonin ($P = 0.003$) or lactate ($P = 0.001$) and high-sensitivity C-reactive protein was statistically significant. There was no difference between procalcitonin and lactate ($P = 0.23$) (Fig. 3).

We assessed the discriminative ability results of single or combined tests to predict positive blood culture (Table 2). With regard to single test items, the sensitivity, specificity,

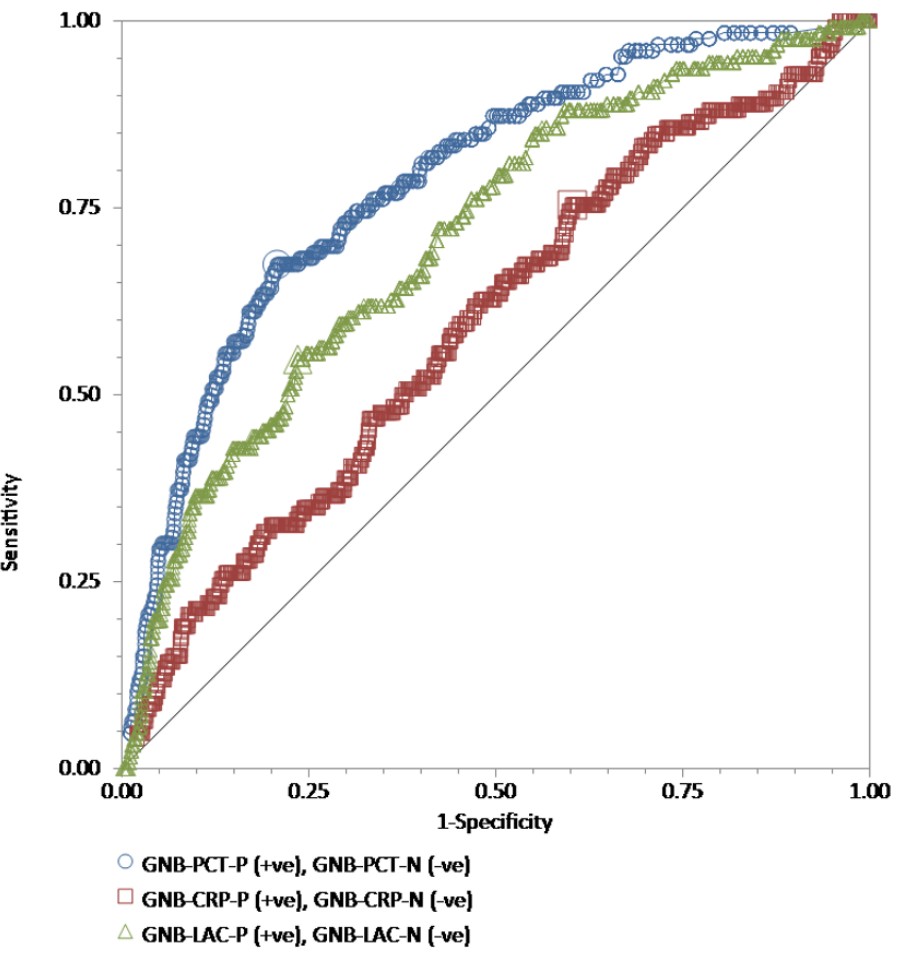

**Figure 2** **The ROC curves for procalcitonin, high-sensitivity C-reactive protein, and lactate results relative to the "gold standard" of blood culture positive for gram-negative bacteria.** The receiver operating characteristic (ROC) curves for procalcitonin (PCT), high-sensitivity C-reactive protein (hs-CRP), and lactate (LAC) results relative to the "gold standard" of blood culture positive for gram-negative bacteria in a total of 815 adult patients admitted to the emergency department. Wilcoxon estimates of the area under the ROC curve are as follows: PCT = 0.79 (95% CI [0.76–0.81]); LAC = 0.71 (95% CI [0.68–0.74]); CRP = 0.60 (95% CI [0.56–0.63]). Pairwise comparisons of ROC curves show that PCT was statistically better than LAC ($P = 0.01$) and hs-CRP ($P < 0.001$), while LAC was statistically better than hs-CRP ($P = 0.003$).

overall accuracy, positive-test likelihood, negative-test likelihood, and diagnostic odds ratio for procalcitonin were 81% (95% CI [74%–86%]), 47% (95% CI [43%–50%]), 54% (95% CI [51%–57%]), 1.51 (95% CI [1.36–1.66]), 0.41 (95% CI [0.31–0.55]), and 3.64 (95% CI [2.46–5.51]), respectively; those for lactate were 63% (95% CI [0.56–0.7]), 63% (95% CI [59%–66%]), 63% (95% CI [60%–66%]), 1.71 (95% CI [1.47–1.96]), 0.58 (95% CI [0.48–0.7]), and 2.93 (95% CI [2.09–4.14]), respectively; and those for high-sensitivity C-reactive protein were 88% (95% CI [82%–92%]), 11% (95% CI [9%–14%]), 28% (95% CI [25%–31%]), 0.99 (95% CI [0.92–1.04]), 1.09 (95% CI [0.71–1.66]), and 0.91 (95% CI [0.55–1.55]; $P = 0.79$), respectively (Table 2). With regard to combined tests, the overall accuracy, positive-test likelihood, negative-test likelihood, and diagnostic odds
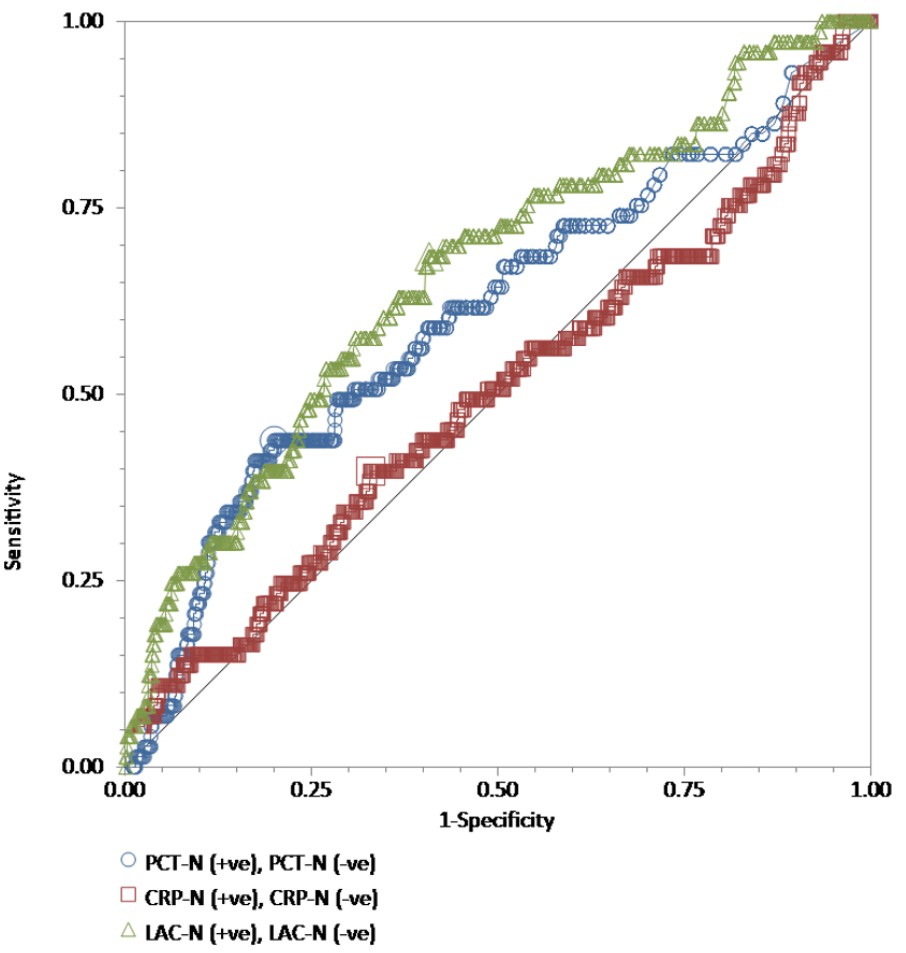

**Figure 3** **The ROC curves for procalcitonin, high-sensitivity C-reactive protein, and lactate results relative to the "gold standard" of blood culture positive for gram-positive bacteria.** The receiver operating characteristic (ROC) curves for procalcitonin (PCT), high-sensitivity C-reactive protein (hs-CRP), and lactate (LAC) results relative to the "gold standard" of blood culture positive for gram-positive bacteria in a total of 762 adult patients admitted to the emergency department. Wilcoxon estimates of the area under the ROC curve are as follows: PCT = 0.61 (95% CI [0.57–0.64]); LAC = 0.66 (95% CI [0.63–0.70]); CRP = 0.50 (95% CI [0.46–0.54]). Pairwise comparisons of ROC curves show that the difference between either PCT ($P = 0.003$) or LAC ($P = 0.001$) and hs-CRP is statistically significant. There is no difference between PCT and LAC ($P = 0.23$).

ratio for procalcitonin and lactate increases were 72% (95% CI [69%–75%]), 2.41 (95% CI [1.98–2.91]), 0.60 (95% CI [0.51–0.70]), and 3.98 (95% CI [2.81–5.63]), respectively. The results of the combined test were slightly better compared to those of the single tests (Table 2).

We assessed the discriminative ability results of single or combined tests to predict positive GNB culture (Table 3). With regard to single test items, the sensitivity, specificity, overall accuracy, positive-test likelihood, negative-test likelihood, and diagnostic odds ratio for procalcitonin were 88% (95% CI [81%–93%]), 47% (95% CI [43%–50%]), 53% (95% CI [50%–56%]), 1.65 (95% CI [1.49–1.81]), 0.26 (95% CI [0.16–0.40]), and 6.44 (95% CI

[3.65–12.15]), respectively; those for lactate were 64% (95% CI [55%–73%]), 63% (95% CI [59%–66%]), 63% (95% CI [60%–66%]), 1.73 (95% CI [1.46–2.02]), 0.57 (95% CI [0.44–0.71]), and 3.04 (95% CI [2.02–4.63]), respectively; and those for high-sensitivity C-reactive protein were 90% (95% CI [84%–95%]), 11% (95% CI [9%–14%]), 23% (95% CI [21%–26%]), 1.02 (95% CI [0.94–1.07]), 0.85 (95% CI [0.48–1.49]), and 1.20 (95% CI [0.62–2.49]; $P = 0.70$), respectively. With regard to combined tests, the sensitivity, specificity, overall accuracy, positive-test likelihood, negative-test likelihood, and diagnostic odds ratio for procalcitonin and lactate increases were 58% (95% CI [49%–67%]), 78% (95% CI [75%–81%]), 75% (95% CI [72%–78%]), 2.64 (95% CI [2.14–3.22]), 0.54 (95% CI [0.43–0.65]), and 4.9 (95% CI [3.24–7.45]); $P < 0.001$), respectively. According to the diagnostic effectiveness (accuracy) results, both procalcitonin and lactate increases provided the highest overall accuracy for the prediction of GNB bacteremia (75%) when compared to the findings of the single tests of procalcitonin (53%), lactate (63%), and high-sensitivity C-reactive protein (23%). However, in terms of the diagnostic odds ratio, the single test of procalcitonin showed better results than the combination (6.44 for procalcitonin alone vs. 4.9 for the combination) (Table 3).

We further assessed the discriminative ability results of single or combined tests to predict positive GPB culture (Table 4). With regard to single test items, the sensitivity, specificity, overall accuracy, positive-test likelihood, negative-test likelihood, and diagnostic odds ratio for procalcitonin at a cutoff of 0.5 ng/mL were 68% (95% CI [57%–79%]), 47% (95% CI [43%–50%]), 49% (95% CI [45%–52%]), 1.28 (95% CI [1.06–1.49]), 0.68 (95% CI [0.47–0.93]), and 1.89 (95% CI [1.11–3.33]; $P = 0.02$), respectively; those for lactate were 63% (95% CI [51%–74%]), 63% (95% CI [60%–66%]), 63% (95% CI [60%–66%]), 1.7 (95% CI [1.36–2.04]), 0.59 (95% CI [0.42–0.78]), and 2.88 (95% CI [1.7–4.94]; $P < 0.001$), respectively; and those for high-sensitivity C-reactive protein were 84% (95% CI [73%–91%]), 11% (95% CI [9%–14%]), 18% (95% CI [15%–21%]), 0.94 (95% CI [0.82–1.02]), 1.47 (95% CI [0.83–2.49]), and 0.64 (95% CI [0.32–1.37]; $P = 0.25$), respectively. With regard to combined tests, the overall accuracy and diagnostic odds ratio for procalcitonin and lactate increases were 75% (95% CI [72%–78%]) and 2.93 (95% CI [1.73–4.96]; $P < 0.001$), respectively. The results of the combined test were slightly better compared to those of the single tests (Table 4).

## Validity analyses

We performed a further analysis to determine whether the discriminative powers of the elevation of procalcitonin ($\geq 0.5$ ng/mL) in a single test and the elevations of procalcitonin and lactate ($\geq 19.8$ mg/dL) in a combined test for predicting positive blood bacterial culture or positive GNB or GPB culture remain in an expanded cohort of similar patients (Tables S1 and S2). The discriminative ability of procalcitonin at a cutoff of 0.5 ng/mL remained in an expanded cohort of 923 patients, with similar results for overall accuracy and diagnostic odds ratio between the single test and the combined test (72% vs. 72% in terms of overall accuracy and 3.98 vs. 3.94 in terms of the diagnostic odds ratio for predicting positive blood culture; 75% vs. 75% in terms of accuracy and 4.90 vs. 5.02 in terms of the diagnostic odds

ratio for predicting positive GNB culture;) 75% vs. 75% in terms of accuracy and 2.93 vs. 2.75 in terms of the diagnostic odds ratio for predicting positive GPB culture (Table S2).

As a single test to predict bacteremia, the procalcitonin test performed even better in a larger expanded cohort of 2,234 adult patients than in the initial cohort of 886 patients in terms of overall accuracy and diagnostic odds ratio, 65% vs. 54% in terms of accuracy and 5.34 vs. 3.64 in terms of the diagnostic odds ratio for predicting blood culture positivity; 65% vs. 53% in terms of accuracy and 10.13 vs. 6.44 in terms of the diagnostic odds ratio for predicting positive GNB culture; and 63% vs. 49% in terms of accuracy and 2.83 vs. 1.89 in terms of the diagnostic odds ratio for predicting positive GPB culture (Table S1).

The predictive performance of high-sensitivity C-reactive protein was consistently poor for predicting positive blood culture, positive GNB culture, and positive GPB culture, according to all three single global diagnostic indicators, namely AUROC analysis, diagnostic effectiveness (accuracy), and diagnostic odds ratio.

## DISCUSSION

In our study, with regard to the discriminative ability to predict bloodstream bacterial infection in adult patients admitted to the ED of a tertiary care medical center, the procalcitonin test was clinically useful to predict positive bacteremia and GNB bloodstream infection. This discriminative power was valid on assessment in a subsequently expanded cohort of 2,234 patients. In terms of a single global indicator, patients with elevated procalcitonin at a cutoff of 0.5 ng/mL have an at least six-fold increased diagnostic odds ratio for GNB bloodstream infection, as compared with normal procalcitonin results. The diagnostic odds ratio does not further improve when elevated lactate at a cutoff of $\geq 19.8$ mg/dL is added to elevated procalcitonin. Elevated procalcitonin can predict positive GPB culture but with a much lower diagnostic odds ratio than that for positive GNB culture. High-sensitivity C-reactive protein performed poorly across all assessments of bloodstream infection, regardless of evaluation using any of the following diagnostic performance indicators: overall accuracy, AUROC analysis, or diagnostic odds ratio. Our data do not support the practice at the ED using high-sensitivity C-reactive protein as a single test at a cutoff of $\geq 0.8$ mg/dL to predict bloodstream bacterial infection.

Using multiple concurrent indicators, such as sensitivity, specificity, and positive-test or negative-test likelihood values, to compare the performance of competing diagnostic tests can be a disadvantage particularly if one test does not outperform the others on all indicators. The diagnostic odds ratio has been proposed as a single indicator of test performance especially in the era of evidence-based practice (Glas et al., 2003). In the stressful setting of the ED involving patient care with a presumptive diagnosis of bacterial bloodstream infection, the identification of the most appropriate test for antibiotic stewardship before the availability of blood culture results and notification of the chance of bacteremia to the patient and family after obtaining the result of the decision test are extremely important. In this study, the procalcitonin test was compared with conventional competing tests, such as high-sensitivity C-reactive protein and lactate, and it was shown that the procalcitonin test has better discrimination ability for blood culture results of

contamination and negative bacterial growth, and its performance has better consistency with blood culture results of positive bacterial growth.

Our study incorporated three different single indicators of test performance for decision-making, namely, overall accuracy, diagnostic odds ratio, and AUROC analysis. Additionally, a negative-test likelihood of <0.6 can help indicate the clinical usefulness of a test. Therefore, one can compare the results of test performance by appreciating the results of the above indicators, and this is a strength of the present study.

We attempted to compare our results with those of other studies. Table S3 presents our analysis of recent studies that reported procalcitonin as a single decision tool and used the diagnostic odds ratio as a single indicator of bloodstream infection. It is noteworthy that for predicting positive GNB culture, a positive procalcitonin test indicates a four- to seven-fold higher chance in terms of the diagnostic odds ratio when compared with a negative test result (diagnostic odds ratio = 4.14 95% CI [2.00–8.58] in the study by Juutilainen et al.; 5.98 95% CI [5.20–6.88] in the study by Oussalah et al. using a cutoff of ≥10 ng/mL; and 6.44 95% CI [3.65–12.15] in our study using a cutoff of ≥0.5 ng/mL) (*Juutilainen et al., 2011*; *Oussalah et al., 2015*). The predictive power of a single procalcitonin test markedly dropped to around two- to four-fold for GPB (diagnostic odds ratio = 1.89 95% CI [1.11–3.33] in our study and 3.64 95% CI [3.11–4.26] in the study by Oussalah et al. using a cutoff of ≥10 ng/mL).

It is worthwhile to look more deeply at the suitable cutoff of procalcitonin level aiming to predict or rule out a positive bacteremia. In a prospective cohort study involving a total of 898 patients fulfilling the systemic inflammatory response syndrome (SIRS) criteria, at the cut-off of 0.1 ng/mL, procalcitonin failed to predict bloodstream infection in 7% of patients (*Hoenigl et al., 2014*). If this cutoff is used to distinguish blood contamination from bloodstream infection due to coagulase-negative Staphylococci, procalcitonin test performed on the same day of blood culture collection had a sensitivity of 100%, and a specificity of 80% for the diagnosis of bloodstream infection (*Schuetz, Mueller & Trampuz, 2007*). Another retrospective single institution study reported that using procalcitonin to predict gram-negative bacteremia, at a cutoff of >3.39 ng/mL, the sensitivity would be 80%, specificity 71%, and the area under the curve 0.73 (*Guo et al., 2015*). A recent Japanese retrospective study involving 1,331 adults with suspicious bloodstream infections derived an optimal cutoff of procalcitonin for discriminating positive blood cultures from negative ones; with the cutoff at 0.9 ng/mL, the sensitivity, specificity, positive predictive value, and negative predictive value were 71.9%, 69.1%, 24.5%, and 94.6%, respectively (*Hattori et al., 2014*). Another South Korean retrospective study involving 300 patients with fever, 58 of them had positive blood culture, when using procalcitonin level at a cut-off value of 0.5 ng/mL, the sensitivity and specificity were 74.2% and 70.1%, respectively. This Korean study also demonstrates that when procalcitonin level is <0.4 ng/mL, it accurately rules out the diagnosis of bacteremia (*Kim et al., 2011*). In a Swiss prospective case-control study accruing 200 hospitalized adults, at a cut-off of 0.5 ng/mL, the sensitivity of procalcitonin was 56%, and the specificity was 83% to discriminate positive blood cultures from negative blood cultures (*Liaudat et al., 2001*). Physicians frequently place the bloodstream infections on the top of the differential diagnoses when a patient presents with the SIRS defined as

two or more abnormalities in temperature, heart rate, respiration, or white blood cell count (*Levy et al., 2003*). A recent study demonstrates that the performance of procalcitonin to predict a bloodstream infection was not affected by the SIRS status. Procalcitonin of <0.1 ng/mL had a negative predictive value of 97.4 and 96.2% for bloodstream infection in the SIRS-negative and SIRS-positive patients, respectively (*Arora et al., 2017*). Using a cutoff at the upper limit of reference range for all three biomarkers, the generalizability of our study results will be more easily applied to the real-world practice in the emergency department.

Interestingly, elevated lactate at a cutoff of ≥19.8 mg/dL is better than elevated procalcitonin in terms of producing a higher diagnostic odds ratio with a two- to three-fold increase in the prediction of positive GPB culture. This association has not been reported in previous studies. In our study, lactate combined with procalcitonin was found to have a better detection capability than high-sensitivity C-reactive protein, and this result may decrease the use of antibiotics in this aspect, when practicing physicians in the ED, who frequently depend on high-sensitivity C-reactive protein to help with the decision to prescribe antibiotics (antibiotic stewardship), make use of decision tools, such as lactate, instead of high-sensitivity C-reactive protein.

From the results of single tests, pairwise test combinations, and blood culture, we noted some important features. First, test combinations may not have better detection ability than a single test. Second, for test combinations, the combination with an increase in a single factor has a higher odds ratio and lower negative-test likelihood than the combination with increases in both factors. Third, elevated lactate and/or procalcitonin show a lower negative-test likelihood and higher odds ratio. Finally, for predicting GNB infection, procalcitonin has better detection capability, and the diagnostic odds ratio does not improve when procalcitonin is used in combination with lactate. To the best of our knowledge, there are few published works investigating whether a combination of tests such as the combination of procalcitonin + lactate or procalcitonin + C-reactive protein or all three of them would outperform procalcitonin as a single test in prediction of the bloodstream infections in adult patients in the setting of emergency department care (*Ljungstrom et al., 2017*). For prediction of bacteremia in adult patients suspected with sepsis admitted to the emergency department at Skaraborg Hospital, Sweden, elevated procalcitonin (AUC = 0.74; 95% CI [0.70–0.78]) performs as good as the composite four biomarkers (procalcitonin, lactate, C-reactive protein and neutrophil and lymphocyte count (NLCR)) (AUC = 0.78; 95% CI [0.74–0.81]) ($P = 0.06$) (*Ljungstrom et al., 2017*). This group of patients had a very high prevalence of verified bacterial infection, at 55.6% (874/1,572).

The present study has some limitations. First, this study only collected data of samples from an academic medical center in central Taiwan, and this could have led to sampling bias. The results from this study may not apply to the ED of other hospitals. Second, during enrollment in this study, the number of cases that fulfilled the enrollment criteria was not large; therefore, this study only classified bacteria as GPB and GNB. Readers are again reminded here that our study assesses predictors of blood culture positivity, but not of sepsis. In further analysis, the sample size should be expanded to increase the credibility of the results. Third, as this study only examined adult emergency patients in a medical center

in central Taiwan, the results may be affected by factors such as prior use of antibiotics by the patient, underlying disease status, conduction of tests simultaneously, improper sterilization during blood sample collection resulting in contamination, and insufficient blood collection resulting in negative results. Subsequent studies could consider including specialty-specific and clinical signs (e.g., temperature, blood pressure, arterial blood gas, leukocyte count, breathing rate, and heart rate).

## CONCLUSIONS

For adult emergency patients, procalcitonin has an acceptable discriminative ability for bacterial blood culture and a better discriminative ability for positive GNB culture when compared with lactate and high-sensitivity C-reactive protein. Although blood culture may be positive, the patient may not have sepsis; therefore, accurate discrimination of contamination is important to ED physicians for treatment purposes. In terms of a single global indicator, patients with elevated procalcitonin at a cutoff of 0.5 ng/mL harbor an at least six-fold increased diagnostic odds ratio for GNB bloodstream infection, when compared with a normal procalcitonin result. The diagnostic odds ratio does not further improve when elevated lactate at a cutoff of 19.8 mg/dL is added to elevated procalcitonin. Elevated procalcitonin can predict positive GPB culture with a much lower diagnostic odds ratio than that for positive GNB culture. High-sensitivity C-reactive protein performed poorly for the prediction of positive bacterial culture. The development of new and effective testing methods will be beneficial for institutions with limited medical resources. Future studies should be performed to determine whether both new and old methods can be used in combination. It is worth mentioning that high-sensitivity C-reactive protein does not show significance for distinguishing blood bacterial culture results of contamination and positivity.

## ACKNOWLEDGEMENTS

We thank Crimson Interactive Pvt. Ltd. (Enago)—http://www.enago.tw for their assistance in manuscript editing.

### Funding

This study was supported by grants to to Chiung–Tsung Lin from the China Medical University Hospital (DMR–99–133), Taichung, Taiwan; and Jorng-Tzong Horng from the Ministry of Science and Technology of Taiwan (MOST 106-2221-E-008-098-MY2). The funders had no role in study design, data collection and analysis, decision to publish, or preparation of the manuscript.

### Grant Disclosures

The following grant information was disclosed by the authors:
China Medical University Hospital: DMR–99–133.
The Ministry of Science and Technology of Taiwan: MOST 106-2221-E-008-098-MY2.

## Competing Interests

The authors declare there are no competing interests.

## Author Contributions

- Chiung-Tsung Lin conceived and designed the experiments, performed the experiments, analyzed the data, contributed reagents/materials/analysis tools, wrote the paper, prepared figures and/or tables.
- Jang-Jih Lu performed the experiments, analyzed the data.
- Yu-Ching Chen analyzed the data, reviewed drafts of the paper.
- Victor C. Kok conceived and designed the experiments, analyzed the data, contributed reagents/materials/analysis tools, wrote the paper, prepared figures and/or tables, reviewed drafts of the paper.
- Jorng-Tzong Horng analyzed the data, reviewed drafts of the paper.

## Human Ethics

The following information was supplied relating to ethical approvals (i.e., approving body and any reference numbers):

The Research Ethics Committee of China Medical University & Hospital, Taichung, Taiwan, recommended and classified the study as an exempt research and granted Ethical approval to carry out the study.

## Data Availability

The raw dataset is provided as a Supplemental File.

## Supplemental Information

Supplemental information for this article can be found online at http://dx.doi.org/10.7717/peerj.4094#supplemental-information.

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
