# Peer review of "Diagnostic value of serum procalcitonin, lactate, and high-sensitivity C-reactive protein for predicting bacteremia in adult patients in the emergency department"

_PeerJ, doi:10.7717/peerj.4094_

## Round 0.1 · original submission · Major Revisions

· Academic Editor

Major Revisions

The two reviewers have provided what I see as very helpful and important comments. I suggest you address all the issues raised. Both reviewers highlight the need to be more robust in the experimental design - including definitions for BC contaminants, that this study is looking at BC positivity vs negativity and not predictors of sepsis, a sensitivity analysis that looks at where only 1 or 2 of the tests have been performed, and rationale for selection of cut-offs for the tests. In terms of validity, it would be helpful to also assess how representative this sample of patients is of the broader group of patients who have had BC taken. Is there something systematically different in these 890 patients compared to the ~41,000 who had BC taken?

Reviewer 1 ·

Basic reporting

1. Title: the title is misleading. The authors have assessed the diagnostic value of several biomarkers commonly used in predicting positive blood cultures, not diagnosing sepsis. The title should reflect this.
2. Abstract: the authors state the “lactate added to procalcitonin helps to improve the performance to differentiate whether blood culture will be positive with either Gram-positive or Gram-negative pathogen”. However, the results presented do not support this conclusion as no biomarker levels are presented. For example, if the patient has elevated lactate and elevated PCT, this information will not help the clinician to differentiate between Gram-positive or Gram-negative bacteraemia. It is rather just an indicator that the patient is more likely have a positive blood culture.
3. Discussion: too few references are used in the Discussion (only three). The results presented in this paper must be considered in context of other people’s work.
4. Abstract: abbreviations should not be used in the abstract.
5. Introduction: the introduction need more detail and relevant references. For example, the previously published results for PCT as a biomarker have been contradictory, which is not mentioned in the introduction.
6. Introduction: references are missing for several claims. For example, line 79-80; line 86-87. Besides, many of the references are relatively old – there have been a lot of publications in this area the latest years.
7. Material and method: no information whether one-sided or two-sided test were used.
8. Material and method: Line 152: the sentence “The p-value in our analysis was set at 0.05” should be reformulated as one cannot set a p-value. Before you perform a test, you decide the significance level (alpha) which is usually set to 0.05.I suggest that you write: “The significance level was set at 5% (p < 0.05).” or “A two-sided p value of <0.05 was considered statistically significant.”
9. Regarding the number of decimals used for reporting p-values both in text and tables: too many decimals used, I suggest that you use two decimals for p-values >0.01; three decimals for p-values between 0.01-0.001; p-values below 0.001 should be reported as p<0.001.
10. Regarding the number of decimals used for reporting the AUROC values: too many decimals used, I suggest that you use two decimals.
11. The 95% CI for AUROC should also be reported, in the text as well as in the figure legends.
12. Results: Interpretations of results and conclusions are presented in the Result section; this belongs to the Discussion section. For example, line 206: “With regards to the single test item, PCT is still a better choice.” and line 209: “Therefore, a combination of PCT or lactate increases is better against blood culture.”, etc.
13. Results: all paragraphs starts with “Table X shows…”. I suggest that the authors report the key results in the text and then referring to the figures and tables in the text.
14. Sometimes the authors use the term odds ratio (OR) and sometimes diagnostic odds ratios (DOR). This is confusing. The authors should stay put to use one of the terms.
15. Table 3: cut-off for lactate is written as 19.8 mg/dl? Confusing as the authors have stated 1.98 mg/dl as cut-off elsewhere in the paper.
16. Table 7: all abbreviations used in the table is not explained in full (e.g., Y, ED, PCT).
17. Table 7: complete references are missing; only the name of the first author is provided.

Experimental design

1. Statistical comparisons between the ROC curves are missing. In order to determine whether there is a significant difference or not you must test for that, not only rank them in order. I suggest that you use DeLong’s test for comparison of two correlated ROC curves (DeLong et al. Comparing the areas under two or more correlated receiver operating characteristic curves: a nonparametric approach. Biometrics. 1988;44(3):837-45).
2. The reasons for selection of the cut-off values for lactate, PCT and CRP should be described. For example, why cut-off values for lactate >1.98 mg/dL and CRP>0.8 mg/dl? These cut-off values have not been used in previous sepsis studies.
3. Material and method: Line 127: information about the blood volume drawn from each patients is missing as well as the number of blood culture bottles incubated for each patient.
4. Material and method: neither information nor reference are given for which bacteria species were interpreted as pathogenic or contaminant.

Validity of the findings

There are many caveats in this paper which limits it validity.
1. Definition of sepsis: this is my major concern about this paper. For several reasons, the results presented in this paper cannot be used in diagnosis of sepsis as claimed, only to predict positive blood cultures. Firstly, I would like to stress that positive blood culture is not the same as sepsis. Many patients suffer from sepsis without having a positive blood culture (only 30-40% of the septic patients have a positive blood culture) and vice versa, many patients can have a positive blood culture without having sepsis. Besides, sepsis may be caused by other pathogens than bacteria. The previous definition (Sepsis-2) of sepsis included proven or suspected infection accompanied by systemic inflammatory response syndrome (SIRS), whereas according to the new sepsis definition (Sepsis-3), sepsis should be defined as life-threatening organ dysfunction caused by a dysregulated host response to infection (Singer et al., 2016. The Third International Consensus Definitions for Sepsis and Septic Shock (Sepsis-3). JAMA 801-810:(8)315). Secondly, the patients enrolled in the study were not selected based on a clinical suspicion of sepsis. Patients were enrolled if all four tests (blood culture, PCT, hs-CRP, lactate) were performed within 24 hours after admission to the hospital – that´s a huge different. No information is provided about the clinical diagnosis nor the reason for taking blood cultures.
2. The paper studied the predictability of positive blood cultures in adult patients, not sepsis. This must be made clear throughout the whole paper.
3. Cut-off values: the selected cut-off values have not been used previous sepsis studies and are in disagreement with the ones generally recommended for clinical practices.
4. The levels for the different biomarkers should be presented as well in the Result section.
5. Sometimes the authors write that they have studied blood cultures positive with Gram-negative bacilli or bacteria/Gram-positive cocci, and sometimes blood cultures positive with Gram-negative /Gram-positive pathogens. This inconsistency is confusing – what have actually been studied? How did the authors evaluate findings like Gram-positive bacilli? Where they considered as negative or contaminations or excluded from the study?
6. Raw data file is supplied, that´s excellent. However, no information is given about the numbers of positive blood culture bottles, no either is any information about the findings classified as contaminants. As the number of positive blood cultures are of importance for deciding whether it’s a pathogen and contaminant it should be provided. What about polymicrobial findings? What about fungal findings? The raw data file also contains some Chinese characters not possible for me to understand. Moreover, units for the biomarkers measurement are missing.
7. Accuracy is another useful performance measure, not included in the paper. It would have been a nice addition to the paper.

Comments for the author

1. Discussion: the studies you compare with have used other cut-off values which makes it difficult to compare. It is more correct to only compare results between studies using same cut-off values.
2. The use of abbreviations: abbreviations should generally be avoided. For the first reference to a term in the text, the term should be used in full with the abbreviation included in brackets. For the remainder of the text the abbreviation should be used. Moreover, I recommend that the authors only used standard abbreviations commonly used (e.g., DOR, CI, PCT, hs-CRP, ROC).
3. Line 238: sepsis is NOT defined as a bloodstream infection.
4. Line 242: “In this study, PCT was compared with conventional competing tests, such as hs-CRP or lactate,…”. I guess that “or” should be changed to “and”?
5. Line 298: the authors write that PCT has a better diagnostic ability for GNB, but in comparison to what?

Reviewer 2 ·

Basic reporting

See general comments

Experimental design

See general comments

Validity of the findings

See general comments

Comments for the author

Thank you for asking to review this paper. This retrospective analysis assessed the utility of lactate, PCT and hsCRP in prediction of blood stream bacteraemia in patients presenting to ED.
General comments:
1. Patients included in this study were treated in 2010 onward, specify the end date.
2. The language used across the manuscript, particularly in the abstract, need significant English editing.
3. Sentences such as “hsCRP should be abandoned ……” should be avoided and deleted.
4. The short title should reflect the biomarker interest. Suggest changing to Biomarkers of sepsis in adult emergency.
5. Why there is no ED physicians included as authors?
6. A sensitivity analysis should have been performed on patients who had one or 2 tests done not just those who have had ALL the 3 tests done.
Specific comments:
1. The mini VIDAS biological range is much lower than 0.5 ng/ml. I thought it goes down to
0.06.

Methods: It is not helpful having too many categories of culture negative. Stick with a definition and just use the total number of what was categorised as negative culture in the analysis.

In the results, graphs and tables.
1. It is very adequate to ONLY separate g-ve and g+ve. No need for ROC for bacilli and cocci. This should apply to all results including the tables. There is little pint dividing cultures into 4 groups in addition to negative, contaminant and others.
2. PCT cut off was chosen at 0.5 ng/ml. Do you have data for PCT cut off at 0.25 or 0.1 ng/ml? In addition is there an upper cut off value for best performance of PCT?
3. Suggest merging tables 1, 2 an 3 into one table.
4. Table 7 is not necessary. This is not a MA. I suggest comparing the results with your data in the discussion section.
Discussion should be restructured to address the following:
1. Start with key results
2. Explain why DOR of combinations is better than single DOR.
3. Compare with previous trials
4. Implication of the above finding for ED practice and sepsis management in general
5. Limitations of the study
6. Next step.
7. Shorter conclusion (should not have any results in the conclusion).

---

## Round 0.2 · Major Revisions

· Academic Editor

Major Revisions

The two reviewers have had another look. There are still some outstanding issues that need to be addressed. I have also include some comments myself below.

Abstract: following the suggested changes in the ‘Discussion’ section, the resulting statement is now confusing. It needs to be reworded for clarity. I suggest removing mention of the Gram positive and Gram negative part. I also suspect Reviewer 1 doesn’t want any abbreviations in the abstract, rather than defining at first mention. I suggest you not use PCT or LAC – just spell these out throughout.

Overall, there still needs to be further English language editing.
Introduction:

What is meant by ‘blood-cultured’ patient? Is this someone who has had a blood culture taken? Or someone with a positive blood culture?
Materials and methods:

Please provide the reason for the difference cut-off values in the manuscript and state that these are different from that used in previous sepsis studies. Please provide discussion about how this impacts upon the generalisability of your study.

You have now provided details of how contaminants were defined, how about pathogens?

Discussion:

Please include direct statements that your study assesses predictors of BC positivity, but not of sepsis.

I agree with Reviewer 1 that the discussion needs to include further work to place your study in the context of the broader literature in this area.

Reviewer 1 ·

Basic reporting

It is still a lack of literature references and context in the Discussion. Only a very few of your results are compared with other peoples work (and here only two other works are cited). The Disucssion must be extended and the obtained results put in a context, i.e., compared with previous studies reporting the performance of the single biomarkers and/or combined biomarkers for predicting positive blood cultures. Reasons for differences/similiarities in findings between studies?

Line 249 and 257: No information about the frequency of positive blood cultures for the expanded cohorts. This is important as the group sizes are very important for the performance characteristics. This information must be added in Method and Material section.

Line 184-186: this sentence is very hard to understand and must be rewritten.

Line 197- : all the data presented in the text here should be presented by tables only.

Line 246: threshold for PCT is not defined here.

Inconsistent use of hs-CRP vs. high-sensitive CRP. This should be looked over throughout the manuscript.

Experimental design

No comments.

Validity of the findings

In the Abstract: You write "Among patients with elevated PCT and/or LAC, the odds for predicting positive blood culture was nearly five-fold higher compared to that in patients with normal levels (diagnostic odds ratio, 4.83; 95% CI, 42 2.84–8.69). The diagnostic odds ratio was nearly eight-fold higher for the prediction of positive gram-negative bacteria (GNB) culture (diagnostic odds ratio, 7.75; 95% CI, 3.56–20.03) and it was nearly three-fold higher for the prediction of positive gram-positive bacteria (GPB) culture (diagnostic odds ratio, 2.87; 95% CI, 1.43–6.41)." I wonder where these figures come from as I don´t recognize the from the results reported in the manuscript? The abstract should only report exactly the same results as in the Results section.

A diagnostic accuracy of only 65% is not acceptable from a clinical point of view (as stated for example in line 275 and line 347). This is considered to be rather poor actually.

It is wrong to say that CRP should be abandoned from the ED; it is still useful as a marker for bacterial infections but is limited by it´s slow kinetics and should be viewed from the timpe point of disease onset.

Reviewer 2 ·

Basic reporting

Nil

Experimental design

Nil

Validity of the findings

Nil

Comments for the author

Thank you for a very improved manuscript.
I have one suggestion to remove the number of patients from the title.

---

## Round 0.3 · accepted · Accept

· Academic Editor

Accept

I have accepted the manuscript. Thank you for working with us in improving the manuscript. I suggest that in the final version, in the abstract you should include the overall findings for bacteremia (i.e., not just gram negative and gram positive). This is lacking from the most recent abstract. Readers will be most interested in the overall prediction for bacteremia, not just gram negative and gram positive. And you should then summarise the specific gram negative and gram positive findings - this is too long in the revised abstract.